



# Influence of convection on stratospheric water vapor in the North American Monsoon region

Wandi Yu[1], Andrew Dessler[1], Mijeong Park[2], and Eric Jensen[2]

[1]Department of Atmospheric Sciences, Texas A&M University, College Station, TX, USA
[2]National Center for Atmospheric Research, Boulder, Colorado, USA

**Correspondence:** Andrew Dessler (adessler@tamu.edu)

**Abstract.**

We quantify the connection between deep convective occurrence and summertime 100-hPa water vapor anomaly over the North America (NA) region and find substantial consistency of their inter-annual variations and that the water vapor mixing ratio over the NA region is up to ~1 ppmv higher when deep convection occurs. We use a Lagrangian trajectory model to
demonstrate that the structure and the location of the NA anticyclone, as well as the tropical upper tropospheric temperature, determine how much contribution the deep convection could make to moistening the lower stratosphere. The deep convection mainly occurs over the Central Plains region, and most of the convectively moistened air is then transported to the center of the NA anticyclone and the anticyclonic structure helps maintain high water vapor content there. Our hypothesis explains both the summer seasonal cycle and inter-annual variability of the convective moistening efficiency in the NA region, and can provide
valuable insight on modeling stratospheric water vapor.

## 1   Introduction

Stratospheric water vapor influences both the climate (Forster and Shine, 1997, 2002; Smith et al., 2001; Solomon et al., 2010; Dessler et al., 2013) and chemistry of the atmosphere (Ramaswamy et al., 1996; Evans et al., 1998; Dvortsov and Solomon,
2001; Shindell, 2001; Stenke and Grewe, 2005). Most of the air reaching 100 hPa has traversed the tropical tropopause layer (TTL), where low temperatures dehydrate the air to typically 3-5 ppmv (Mote et al., 1996; Sherwood and Dessler, 2000; Randel et al., 2004; Fueglistaler, 2005; Fueglistaler et al., 2009). However, over the Asian monsoon and North American monsoon, higher water vapor mixing ratios, sometimes as high as  12 ppmv, are observed, indicating that the air did not go through the TTL or is moistened further after leaving the TTL (Anderson et al., 2012; Schwartz et al., 2013; Randel et al., 2015; Smith
et al., 2017).

Convection penetrating the tropopause plays a potentially important role in moistening the stratosphere (Dessler and Sherwood, 2004; Dessler et al., 2007; Hanisco et al., 2007; Ueyama et al., 2015, 2018). Previous case studies have shown that





deep convection over North America (NA) can reach the lowermost stratosphere (between the local tropopause and the 380 K isentropic surface), and can even enter the stratospheric overworld (above 380 K), thereby bringing a high water vapor content to the stratosphere (Hanisco et al., 2007; Herman et al., 2017; Smith et al., 2017). However, previous studies on the long-term behavior of NA stratospheric water vapor and deep convection conclude that there is little connection (Randel et al., 2015; Sun and Huang, 2015; Kim et al., 2018). Model simulations, using prescribed global satellite-derived deep convection, also cannot reproduce a high water vapor mixing ratio over NA (Ueyama et al., 2018; Wang et al., 2019) .

One crucial issue in these analyses is how high convection extends into the stratosphere. Previous studies based on the infrared satellite cloud-top height measurements are low biased (Ueyama et al., 2018; Schoeberl et al., 2019) owing to the isothermal nature of the lowermost stratosphere as well as the fact that convective clouds rapidly sublimate in the dry stratosphere and therefore may be missed in observations from polar orbiting satellites. A better option for estimating cloud-top height over NA is ground-based radar (Liu and Liu, 2016; Cooney et al., 2018). The hourly interval GridRad data, derived from NEXRAD radar data, captures convective overshooting events over most parts of the contiguous United States (Solomon et al., 2016; Cooney et al., 2018), sometimes extending as high as 22 km.

In this study, we focus on the NA region and use the GridRad data as a convective proxy to study the impact from deep convection on stratospheric water vapor. In the following analysis, we will first show the relationship between the inter-annual variability of the water vapor and deep convection over the NA region. Then we use a back trajectory model to illustrate the processes that influence whereby deep convection moistens the NA stratosphere, from the perspectives of spatial distribution, seasonal cycle, and inter-annual variability.

## 2 Data and methods

### 2.1 Data

The Microwave Limb Sounder (MLS) on board the NASA Aura satellite (Waters et al., 2006) has provided high-quality daily near-global observations of water vapor in the upper troposphere and stratosphere since August 2004. In our research, we use the version 4.2 level 2 product from 2005 to 2016 and focus on water vapor at 100 hPa over the NA region (25°N - 50°N, 70°W- 130°W). MLS makes roughly 2500 observations over this region every month and we bin and average these MLS data into a 4°latitude by 8°longitude grid after applying quality control (Livesey et al., 2018).

The GridRad dataset archives national NEXRAD WSR-88D radar data (version 3.1). Gridrad v3.1 has horizontal resolution 0.02°longitude and 0.02°latitude, and covers 25°N to 49°N, and 70°W to 105°W, or most parts of the contiguous United States (Homeyer, 2014; Homeyer and Kumjian, 2015; Solomon et al., 2016; Cooney et al., 2018). The vertical resolution of the GridRad data is 1 km, from 1 to 24 km above sea level and it has a temporal resolution of one hour. In our analysis, we identify deep convection as observations of reflectivity $Z_h$ over 10 dBz (Cooney et al., 2018).

Finally, we bin the occurrence of convective cloud into a 1°latitude by 1°longitude (horizontal) by 1 km (vertical) grid box, dividing the number of observations that encounters convection by the total number of observations in each grid box.





Temperature, pressure, and tropopause information comes from the European Centre for Medium-Range Weather Forecast (ECMWF) Re-Analysis (ERA)-Interim (ERAi) archived 6-hourly model fields (Dee et al., 2011). The ERAi data is gridded into a 1°x 1°horizontal grid with 37 vertical layers. To identify the tropopause, we linearly interpolate ERAi temperature into 2 hPa pressure intervals and define the lowest level where the lapse rate is less than 2 K/km to be the tropopause (World Meteorological Organization, 1957).

## 2.2    Back trajectory model

To identify water vapor observations that had previously encountered convection, we use back trajectory analyses. These use temperature, wind, and diabatic heating rate from the ERAi to drive the TRAJ3D Bowman trajectory code (Bowman, 1993; Bowman and Carrie, 2002). Vertical velocities in isentropic coordinates are computed from 6-hourly average diabatic heating rates. Horizontal velocities come from 6-hourly instantaneous two-dimensional horizontal wind fields. Several previous studies
have successfully identified convectively influenced air masses based on trajectory model driven by reanalysis data (Wright et al., 2011; Bergman et al., 2012, 2013; Smith et al., 2017).

In these experiments, we initialize air parcels at and surrounding the MLS observations made during June -August (JJA) over 2005-2016. The initiation of parcel positions follows the same strategy as Smith et al. (2017): We initiate a cluster of 27 parcels on a 3 x 3 x 3 gird surrounding each MLS measurement ($\pm0.25°$latitude, $\pm0.25°$longitude, and $\pm5$ K potential temperature
around and at 100 hPa). We advect the parcels back 5 days and record latitude, longitude, potential temperature, pressure, and temperature every hour.

After performing the back trajectory model, we then divide the trajectories into two groups, depending whether the parcels encountered deep convection along the path or not. The definition of encountering deep convection is if the trajectory is within $\pm0.25°$latitude and $\pm0.25°$longitude of a GridRad deep convection observation and when the parcel is below the convection top
and above the local tropopause at the time the observation is made. If at least one parcel in the cluster of parcels encounters deep convection, the corresponding observation is categorized as encountering convection; otherwise, it is in the 'no-convection' group.

## 3    Relationship between area-average water vapor and convection

In Fig. 1.a, we show the time series of 5-day average 100-hPa water vapor anomaly and 10-day average convective occurrence,
both averaged over the NA region. We have subtracted the zonal mean value (based on the JJA mean from 70°W to 130°W at each degree of latitude) from NA the 100-hPa water vapor content before normalizing, allowing us to focus on the variability in the NA region relative to the zonal average value.

Overall, there is a high correlation between the water vapor anomaly and convective occurrence, which suggests that deep convection moistens the stratosphere. There are exceptions, however — *e.g.*, during June 2010 (marked by the arrow), the
water vapor anomaly is high, despite deep convection being relatively infrequent.



To make this correlation clearer, Figs. 1b-d show scatter plots of the 5-day water vapor anomaly and deep convective occurrence over NA in June, July, and August. We find that deep convection increases 100-hPa NA stratospheric water vapor by up to ~1 ppmv. The slope of the linear fit in Figs. 1.b-d represents the moistening efficiency, which is defined as the amount of water vapor content added per unit of deep convective occurrence in the stated month added. This moistening efficiency is significantly lower during June than July and August, which we will be explained in the next section.

One must be careful not to confuse correlation with causality. We therefore use the back trajectory model to demonstrate the causal relationship implied in Fig. 1. As discussed in section 2.2, we divided the 100 hPa MLS observations into two groups depending on whether they encountered the deep convection during the 5-day back trajectory or not. Fig. 1a shows the convective influence, the fraction of MLS observations that encountered convection and Fig. 2 shows the probability density function (PDF) of water vapor mixing ratio during June, July, and August 2005-2016 in the two groups.

We see that the no-convection group has a similar PDF shape in June, July, and August, with peak values around 4 ppmv. For the MLS measurements that encountered convection, the peak of the PDF is 5-6 ppmv during July and August, and 4-5 ppmv during June, 0.37 ppmv, 0.62 ppmv, and 0.69 ppmv higher than the no-convection group during June, July, and August, respectively.

Our work in this section establishes that deep convection is increasing water vapor over the NA region. However, three questions remain to be answered: First, can deep convection explain the spatial distribution of the water vapor anomaly? Second, why is the convection more effective in July and August than in June? Third, why is there inter-annual variability in the effectiveness of moistening (for example, June 2010 *vs.* June 2011)? These are three key questions we answer in the following sections.

## 4   Differences between June, July, and August

From June to August every year, the water vapor mixing ratio over NA shows positive anomalies relative to the zonal mean (Fig. 3a-c). Deep convection also frequently occurs during boreal summer, especially over the central US (contours in Fig.3 a-c, see also Cooney et al. (2018)). However, there is a discrepancy between the spatial distribution of the water vapor anomaly and deep convective occurrence: The deep convection occurs mainly over the Central Plains region, centered around 40°N. Large positive water vapor anomalies are observed over a broader longitude range south of 40°N latitude.

Our back trajectory calculations show that regions with high convective influence ratios (Fig. 3d-f) tend to be collocated with large positive water vapor anomalies (Fig. 3 a-c, also black dashed contours in Fig. 3d-f). Here, we define the convective influence ratio as the number of MLS observations in each grid box that encountered deep convection during the past 5 days, divided by the total number of MLS observations in that grid box. This collocation suggests that the pattern of enhanced water vapor seen by MLS can be explained by frequent convection. It is worth mentioning that previous studies have also suggested that the water vapor maximum over the NA region cannot be reproduced without the inclusion of convection in the model (Ueyama et al., 2018, their Fig. 3c); (Wang et al., 2019, their Fig. 2f).





Also shown in Fig. 3 (panels g, h, and i) is the NA residence time, as calculated by back trajectory analyses. We initialize the parcel evenly on a 1°x 1°grid over NA (25°N - 50°N, 70°W- 130°W) every hour during each month, and track their positions

back 10 days. We then calculate the time from when it entered the NA region to the initialization point, and then grid the time of these parcels according to their location of initiation. The NA residence time indicates how long the air parcels in each grid box have been in the NA region. These figures show that parcels at the center of the monsoon have the longest history over NA a week or longer.

There is also a similarity between the distribution of the time spent over NA (Fig. 3g-i) and the convective influence ratio, indicating that the monsoon circulation tends to hold air that has flowed over convection in the NA region. This provides an

explanation for the observations in Figs. 3a-c showing that convection tends to be located north of the 100-hPa water vapor maximum.

The monsoon dynamics are also an essential factor in the seasonal cycle of water vapor anomaly. Here, seasonal cycle refers to the increase in water vapor mixing ratio through the summer, from June to July to August. There are two reasons for this.

First, the North American monsoon anticyclone (NAMA) forms in June and enlarges and becomes stable during July and August (Clapp et al., 2019). This leads to increases in the average NA residence time from 3.4 to 4.5 to 4.9 days from June to August. This increases the convective influence (the fraction of MLS observations that encountered convection), with values in June, July, and August of 0.038, 0.081, and 0.091, respectively. What is happening here is that, later in the summer, the convectively moistened air is more likely to be confined within the NA region instead of being transported downwind by the

zonal mean flow. As a result, the deep convection can moisten the NA region more efficiently late in the summer compared to early in the summer.

The second reason is also connected to the changing dynamics during June, July, and August. Parcels tend to travel to lower latitudes during June (Fig. 4a), which leads them to experience colder temperatures at 100 hPa (Fig. reffig4b). This means that convectively moistened air experiences subsequent dehydration more frequently in June than in later months (Randel et al.,

140 2015).

The PDFs of the minimum saturation water vapor mixing ratio, which limits the amount of water in the parcel, indicates that parcels in June tend to have lower values (Fig. 4c). If we choose a minimum saturation water vapor mixing ratio of 5 ppmv as a threshold of effective moistening (stratospheric water vapor mixing ratio commonly won't exceed this value), then 88.0%, 97.8%, and 97.5% of the observations that encountered deep convection are effectively moistened in June, July and

August, respectively. We calculate the effective convective influence ratio by dividing the number of convectively moistened observations that have a minimum saturation water vapor mixing ratio over 5 ppmv by the total number of observations. The effective convective influence ratio is 0.039, 0.082, and 0.092 during June, July, and August, respectively.

## 5  Interannual variability

Figure 1a shows times series for water vapor, deep convection, and the convective influence ratio for June-August of 2005-

2016. The correlation coefficient between water vapor and the effective convective influence ratio time series is 0.74, and 0.60



between water vapor and deep convection. Despite the high correlation between these time series, there are also clear outliers *e.g.*, June 2010 has lower convection, but much higher water vapor than June 2011. In this section, we compare these two months to illustrate the factors that contribute to the interannual variability of convective moistening.

During June 2010, a stable anticyclone forms over the eastern NA region from June 11 to June 30 (Fig. 5, upper panel). In June 2011, the monsoon anticyclone is located further south, and the NA region is dominated by the westerly winds. Because of the difference in locations of the monsoon anticyclone, the parcels influenced by convection experience different pathways. In June 2010, 23% of the convectively influenced parcels travel to the tropics (20°N-20°S) in 5 days, while in June 2011, 65% do (Fig. 6a). This means that parcels influenced by convection in June 2011 on average experience colder temperatures (Fig. 6b). Finally, the tropics were slightly cooler during June 2011 compared to June 2010, which further contributed to lower water vapor in NA. The net result of these differences is that convectively influenced parcels retain more water vapor in June 2010 than in June 2011 (Fig. 6c) due to differing monsoon dynamics. Thus, monsoon dynamics variability plays a significant role in the generating inter-annual variability of stratospheric water vapor in the NA region.

## 6   Conclusions

In this study, we investigated the contribution of convection to the water vapor in the North American (NA) monsoon region, including the seasonal cycle and interannual variation of convective contributions during the Northern Hemisphere summer. We have shown that the deep convection moistens the lower stratosphere, adding on up to ~1 ppmv to the summertime NA water vapor at 100 hPa based on the observations from MLS.

We have also shown that it is not the amount of convection alone that determines the impact on water vapor — NA monsoon dynamics also play a role. We note that the location of deep convection is not collocated with the maximum water vapor in NA, and this is due to high water vapor content being transported downstream by the monsoon circulation. The maximum water vapor content appears near the center of the NA anticyclone.

We also analyzed the seasonal cycle of convective influence. During June, the NA monsoon circulation is located further south than during July and August, so air influenced by convection during June experiences colder temperatures while travel-ing to the tropics. Subsequent dehydration reduces the net moistening from convection during June compared to those other months. Variations in the monsoon dynamics can also lead to interannual variations in convective moistening through a similar mechanism. We compare June 2010 and June 2011 and show that a more northerly monsoon circulation in June 2010 leads to convectively influenced air encountering warmer temperatures, leading to higher water vapor than in June 2011.

Our use of GridRad data as a source of convection is a limitation in our analysis because it only covers the continental US. Much of the monsoonal deep convection also occurs over the Gulf of Mexico (Clapp et al., 2019), out of range of the NEXRAD stations. Future studies including convective data with a larger spatial extent may find that the deep convection over the Gulf of Mexico influences NA stratospheric water vapor, but we do not expect this will conflict with the main conclusions from our paper.



*Data availability.* Deep convection data from GridRad data: http://gridrad.org/data.html (Homeyer, 2014)

Water vapor observed by MLS: https://mls.jpl.nasa.gov/products/h2o_product.php (Waters et al., 2006)

Wind, Temperature, and heating rate are from ERAinterm: https://www.ecmwf.int/en/forecasts/datasets/ (Dee et al., 2011)

*Author contributions.* WY performed the analysis and wrote the original draft. AED provided guidance on the outline of the research, and also edit the paper. MP and EJ provided guidance and discussion on section 3&4 and participate in editing.

*Competing interests.* The authors declare that they have no conflict of interest.

*Acknowledgements.* We thank William J. Randel and Laura Pan for their helpful discussion. This work was supported by NASA grants

80NSSC18K0134 and 80NSSC19K0757. This material is also based upon work supported by the National Center for Atmospheric Research, which is a major facility sponsored by the National Science Foundation under Cooperative Agreement No. 1852977. Any opinions, findings and conclusions or recommendations expressed in this material do not necessarily reflect the views of the National Science Foundation.





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





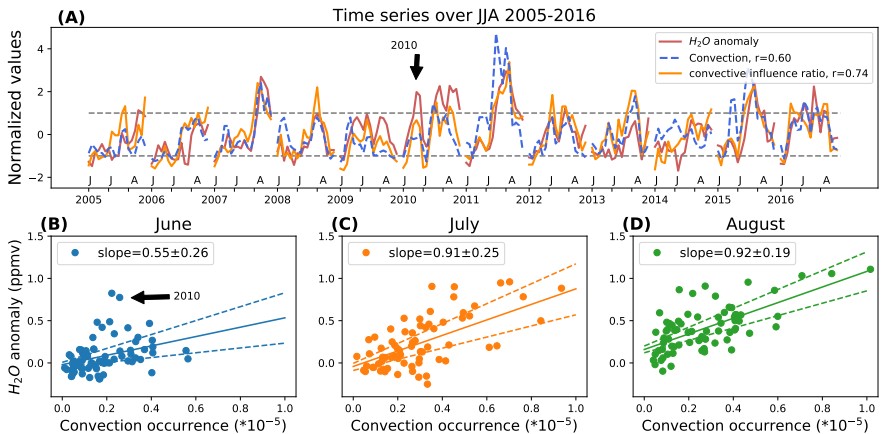

**Figure 1.** (A) Time series of normalized 100-hPa (red line) MLS water vapor anomaly (zonal mean removed), (blue line) convective occurrence from GridRad, and (orange line) the convective influence ratio in the back trajectory experiments. All data are 5-day averages over the NA region ($25°N$-$50°N$, $70°W$-$130°W$) during June, July, August 2005-2016. For the convective frequency, we use linear interpolation to estimate the value at 100 hPa. The convective influence ratio is the fraction of the MLS observations that encounter deep convection, as determined by the back trajectory calculations. (B-D) Joint distribution of convection and water vapor time series during 2005-2016 divided into (B) June, (C) July, and (D) August. Solid lines show the linear fit, and dashed lines show the 95% significant level margin of error of the slope bar (accounting for auto-correlation). To account for the time for the water vapor to spread out, each data point is a 10-day average of convection, with water vapor averaged over the last five days of the averaging period for convection.

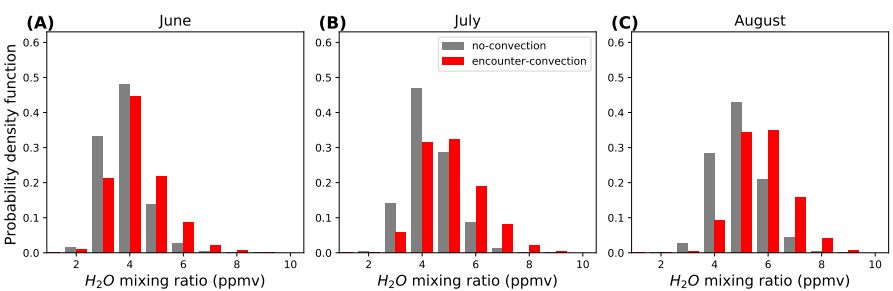

**Figure 2.** Probability density function of MLS water vapor observations at 100 hPa averaged over NA ($25°N$-$50°N$, $70°W$-$130°W$) in (A) June, (B) July, and (C) August 2005-2016. The observations are divided into two groups: (red) those whose back trajectory encounters convection and (grey) those that do not.





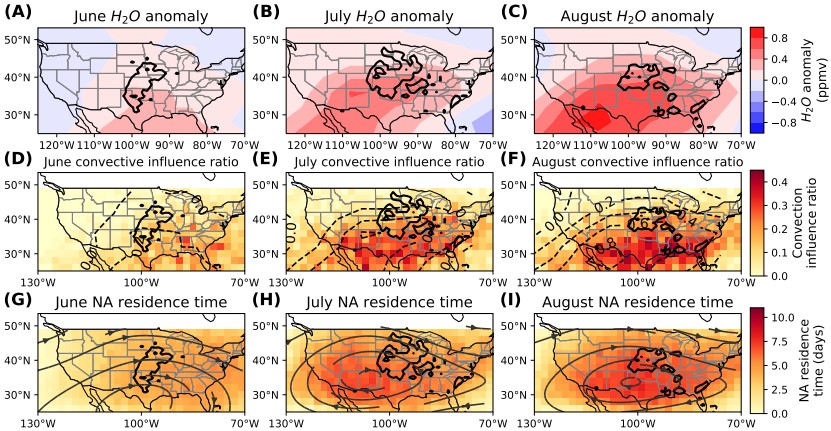

**Figure 3.** (Top row) Geographical distribution of the MLS 100-hPa water vapor anomaly (after removal of the zonal mean), averaged over 2005-2016 in (A) June, (B) July, and (C) August. The black contour in each panel (repeated in each row) is the $10^{-5}$ contour of GridRad convective occurrence, averaged over that month. (Middle row) Geographical distribution of the convection influence ratio over the NA region during (D) June, (E) July and (F) August 2005-2016; (The black dashed contours in each panel are water vapor anomaly contours matching the shading in the corresponding upper panel). (Bottom row) Geographical distribution of the parcel time spent over the NA region during (G) June, (H) July, and (I) August. (The stream lines are horizontal velocities interpolated onto 100 hPa using the cubic spline method from ERAi data averaged over the same period.

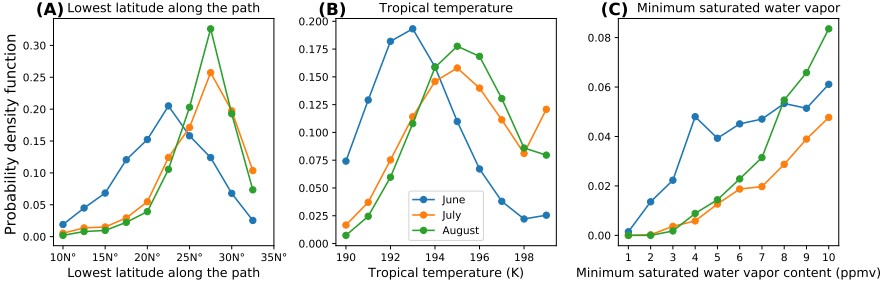

**Figure 4.** . (A) PDF of lowest latitude; (B) PDF of hourly ERAi daily temperature (K) interpolated to the location of the parcels; (C) PDF of minimum saturated water vapor (ppmv). The minimum water saturated water vapor mixing ratio and the lowest latitude are the minimum values along the path after the parcels encounter deep convection in the back trajectory model and prior to being observed by MLS.



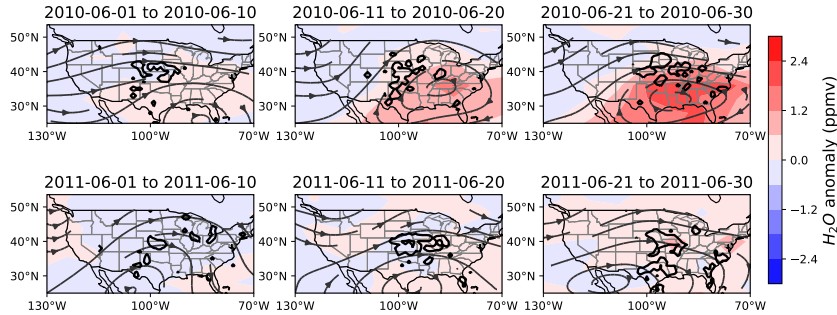

**Figure 5.** Ten day average water vapor anomaly (ppmv) at 100 hPa (after removing the zonal mean) during (top) June 2010 and (bottom) June 2011. Thick black contours are the $10^{-5}$ contours of GridRad deep convection, and stream lines are horizontal velocity at 100 hPa obtained from ERAi averaged over the same period.

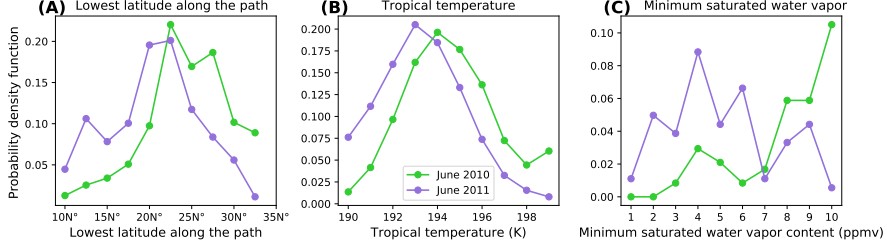

**Figure 6.** PDF of (A) lowest latitude along the path in the back trajectory experiments (B) mean tropical temperature (K) obtained from ERAi ($0°$-$20°$N, $70°$W-$130°$W), and (C) minimum saturation water vapor mixing ratio (ppmv) in the back trajectory experiment results. Minimum saturation water vapor mixing ratios and lowest latitude are the minimum values along the path after the parcels encounter deep convection in the back trajectory model and prior to being observed by MLS.