# Peer review of "Influence of convection on stratospheric water vapor in the North American Monsoon region"

_Atmospheric Chemistry and Physics, 2020_

## Referee Comment (RC1) · Anonymous Referee #1 · 8 Jun 2020

This is an interesting paper and certainly should be published, however the major claim in the paper has not been supported by the data. It is not that the interpretation is implausible, but insufficient evidence has been provided. The paper is certainly of sufficient interest even if this claim is withdrawn.

From my understanding of the manuscript, I do not believe that the primary conclusion "... most of the convectively moistened air is then transported to the center of the NA anticyclone and the anticyclonic structure helps maintain high water vapor content there." is necessarily supported by the evidence provided. What is shown is that there is an offset between the region that corresponds to the authors chosen metric of deep convection, and the region of maximum H2O anomaly as measured by MLS.

Even if one accepts the chosen convective proxy, there seems to be another plausible explanation for this geographic mismatch. Air simply resides in the Southern US region for a longer period before being transported downwind to other longitudes, and therefore has greater convective moisture efficiency. Thus, it is not clear to me to that transport from higher latitudes is required. The claim that can easily be made is that there is a strong correlation between air that has a long residence time over NA and air with a large H2O anomaly.

I am also skeptical about the relevance of the fact that in June the back trajectories travel through very cold regions. These are back trajectories, so these low temperatures are influencing the parcels before they are moistened by the deep convection. Perhaps the authors are suggesting that these parcels are moistened over the US, travel to the tropics, and then come back to the US, but I would have thought that such a trajectory path would only be followed by only a very small fraction of parcels and the authors have provided no evidence to the contrary.

Additional comments:

Abstract line 8– I'm not sure what the "hypothesis" is, nor is there a particular need to mention or have one.

Line 52 – The convective radar data play a fundamental role in this paper. Although the reference to Cooney et al. is good, please devote a few lines to describing the reflectivity observations and what they mean. For instance, is a reflectivity Zn over 10 dBz a well-accepted value for tropopause overshooting convection?

Figure 1 – I understand that in panel 'A' different very different quantities are being plotted, but just putting a y-axis that says "normalized" is unacceptable. There needs to be some way for the reader to connect the plotted value with a physical quantity (ppmv, fractional occurrence of convection, etc.). Also, the two reddish lines in panel 'A' (apparently one is orange) are very difficult to distinguish.

Figure 3 – This is an interesting figure, and the fact that the convective influence ratio looks not dissimilar from the MLS H2O anomaly is very interesting. The black line single value GridRad contour is useful because it makes it easier to see the offset between the maximum convection and the maximum H2O anomaly, and if this specific convection contour represented all convection, then one would be forced into the conclusion presented by the authors, i.e. that high H2O air is being transported from these regions to the lower latitude regions. But there is nothing special about the specific convection level contour that the authors have chosen. While some of the moist air may have been brought down from the North, some of the moistening in the Southern US is almost certainly caused by local convection in this region. This figure therefore include a full row showing a color contour of the convective occurrence by month throughout the US.

[Figure]

---

## Referee Comment (RC2) · Anonymous Referee #2 · 16 Jun 2020

The paper presents a hypothesis that deep overshooting convection over the great plains of north America moistens the lower stratosphere and is transported and trapped in the North American anticyclone. they use trajectory calculations to see if water vapor measurements from MLS encountered convection or not. They show that during July and August that the difference between convective and non convective trajectories is nearly 1 ppmv. In June it is much less. The take away message i get from this is that the establishment of the North American anticyclone is the dominant factor for having high lower stratospheric water vapor over North America. It completely dominates convective activity in the Junes of 2010 and 2011 where the former year is high with low NA convection and low the following year despite having more NA convection. i have difficulty believing that in 2010 before the anticyclone is set up that the convectively

moistened air moves to the colder tropics where it gets freezedried first. It seems from looking at the figure 3 and figure 5 that the horizontal tape recorder signature of water transport is playing a big role here too. i certainly agree that deep convection over the NA plains is adding water but it might be a small perturbation on top of the large scale transport coming up from the tropics that is also becoming more moist during the summer months. I think this could be disentangled with some modelling studies where one could artificially hold the tropical tropopause temperature constant all year thus removing the tape recorder signatures from the tropics and seeing what NA enhancements occur just due to local convection.

minor recommendations page 1 line 18 replace sometimes as high as with exceeding ... MLS has seen higher values as has Anderson.

page 3 line 81 from NA the 100 to from the NA 100...

Throughout the manuscript lower case letters a, b, .. are used to refer to panels in the figures but the figures use upper case letters A, B, ... Please make this consistent.

page 5 line 135 I would write that sentence as As a result moistening from deep convection becomes less diluted by zonal mean flow later in the summer.

---

## Author Comment (AC1) · 30 Jun 2020

We thank the reviewer for their useful comments on our manuscript. Below we provide detailed response to your comments. In the following, the comments of the reviewer are presented in *italic blue*. Our responses are in normal black font. Changes to the text are in red.

**Comment 1:**

From my understanding of the manuscript, I do not believe that the primary conclusion "... most of the convectively moistened air is then transported to the center of the NA anticyclone and the anticyclonic structure helps maintain high water vapor content there." is necessarily supported by the evidence provided. What is shown is that there is an offset between the region that corresponds to the authors chosen metric of deep convection, and the region of maximum H2O anomaly as measured by MLS.

Even if one accepts the chosen convective proxy, there seems to be another plausible explanation for this geographic mismatch. Air simply resides in the Southern US region for a longer period before being transported downwind to other longitudes, and therefore has greater convective moisture efficiency. Thus, it is not clear to me to that transport from higher latitudes is required. The claim that can easily be made is that there is a strong correlation between air that has a long residence time over NA and air with a large H2O anomaly.

We agree that we need to show more analysis to support the conclusion, so we will add a new row to Fig. 3 in the paper; the revised figure is shown below. To produce the new row (the fourth one), we take MLS observations that our back-trajectory analysis shows were convectively enhanced and show the location where those trajectories encountered convection. We find that most of the convective encounters happened over the regions where GridRad data show deep convection is most frequently occurring. This supports our conclusion that the offset between convection and high water vapor is due to transport.

We acknowledge that we cannot rule out a contribution from unobserved convection in regions of high water vapor, but we see no evidence to support that.

In the revised text, we will add this row to Fig. 3 in the paper and insert a paragraph after line 118:

We identify the locations where parcels encounter deep convection in the back trajectories and grid the number of trajectories encountering convection into 2° x 2° boxes (Fig. 3J-L). Most of the locations of convective encounters occur over the region where NEXRAD data show deep convection frequently occurs, e.g., over the Central Plain region, and over Florida during August. The geographical distribution of convective encounters does not match with the convective influence ratio, indicating that convective moisture is transported to the region of high water vapor by the dynamics of the monsoon.

Comment 2:

I am also skeptical about the relevance of the fact that in June the back trajectories travel through very cold regions. These are back trajectories, so these low temperatures are influencing the parcels before they are moistened by the deep convection. Perhaps the authors are suggesting that these parcels are moistened over the US,

**travel to the tropics, and then come back to the US, but I would have thought that such a trajectory path would only be followed by only a very small fraction of parcels and the authors have provided no evidence to the contrary.**

Yes, our argument is that parcels encounter deep convection over the US, travel to the (sub)tropics, are dehydrated along the way, and then travel back over the US. However, as the reviewer correctly suggested, only a minority of parcels take this path. If we regard 20°N-20°S as the tropics, then 33%, 13%, and 9% of the convectively influenced parcels follow this pathway during June, July, and August over 2005-2016, respectively. This reflects the fact that the monsoon circulation is weaker during June, and so a small but important fraction of parcels can follow this path; during July and August, when the monsoon is better established, this pathway is nearly shut off.

As we argue in the paper, year-to-year variations around these average values are responsible for observable variations in water vapor. During June 2011, 44% of the convectively influenced parcels traveled to the tropics, while during June 2010, only 20%. Per the Clausius-Claperyon equation, a 1-K change in minimum temperature will change water vapor by 20% at TTL temperatures. So, 20% of the parcels in 2011 experiencing temperatures a few degrees colder than those in 2010 (as suggested by Fig. 6 of the original paper) can change the water vapor over the monsoon region by the amount observed.

However, we acknowledge that our text could be clearer. We will make the following change to the text:

Line 137 - The second reason is also connected to the changing dynamics during June, July, and August. Parcels tend to travel to lower latitudes during June (Fig. 4a). 33% of the convectively moistened parcels travels to the tropics (20°N-20°S) during June, and 13% during July and 9% during August. Traveling to the tropics leads them to experience colder temperatures at 100 hPa (Fig. 4b). As a result, the median of the water vapor mixing ratio of the parcels that stays in the mid-latitudes is 5.98 ppmv, while it is 5.36 ppmv for those parcels that travels to the tropics. This means that convectively moistened air experiences subsequent dehydration more frequently in June than in later months (Randel et al., 2015).

**Additional comment 1:**

Abstract line 8– I'm not sure what the "hypothesis" is, nor is there a particular need to mention or have one.

Changes have been made in the text: We have replaced "Our hypothesis" with "This".

**Additional comment 2:**

Line 52 – The convective radar data play a fundamental role in this paper. Although the reference to Cooney et al. is good, please devote a few lines to describing the reflectivity observations and what they mean. For instance, is a reflectivity Zn over 10 dBz a well-accepted value for tropopause overshooting convection?

Changes have been made in the text:

Line 52 - Cooney et al. (2018) used GridRad data to calculate the deep convective echo top, and found out a highest level that the reflectivity over 10 dBz is a representative threshold that balances the sensitivity and noise. In our analysis, we also use this strategy and identify deep convection as observations of reflectivity over 10 dBz.

**Additional comment 3:**

Figure 1 - I understand that in panel 'A' different very different quantities are being plotted, but just putting a y-axis that says "normalized" is unacceptable. There needs to be some way for the reader to connect the plotted value with a physical quantity (ppmv, fractional occurrence of convection, etc.). Also, the two reddish lines in panel 'A' (apparently one is orange) are very difficult to distinguish.

Fig. 1A has been revised. We now have multiple y-axes, with each showing the values of each physical quantity. A green line replaces the orange line, which should improve the figures clarity. The revised Fig. 1 is shown below.

**Additional comment 4:**

Figure 3 – This is an interesting figure, and the fact that the convective influence ratio looks not dissimilar from the MLS H2O anomaly is very interesting. The black line single value GridRad contour is useful because it makes it easier to see the offset between the maximum convection and the maximum H2O anomaly, and if this specific convection contour represented all convection, then one would be forced into the conclusion presented by the authors, i.e. that high H2O air is being transported from these regions to the lower latitude regions. But there is nothing special about the specific convection level contour that the authors have chosen. While some of the moist air may have been brought down from the North, some of the moistening in the Southern US is almost certainly caused by local convection in this region. This figure therefore include a full row showing a color contour of the convective occurrence by month throughout the US.

We agree that there is nothing special about the 1e-5 contour. We have therefore taken the reviewer's suggestion and added a row with the maps of convective occurrence over the NA.

Changes have been made in the text:

Line 106: From June to August every year, deep convection frequently occurs during boreal summer, especially over the central US (Fig.3A-C, see also Cooney et al. (2018)). The water vapor mixing ratio over NA also shows positive anomalies relative to the zonal mean (Fig. 3D-F). However, there is a discrepancy between the spatial distribution of the water vapor anomaly and deep convective occurrence: The deep convection occurs mainly over the Central Plains region, centered around 40°N. Large positive water vapor anomalies are observed over a broader longitude range south of 40°N latitude.

Figure 1. (A) Time series of normalized 100-hPa (red line) MLS water vapor anomaly (zonal mean removed), (blue line) convective occurrence from GridRad, and (green line) the convective influence ratio in the back trajectory experiments. All data are 5-day averages over the NA region (25°N-50°N, 70°W-130°W) during June, July, August 2005-2016. For the convective frequency, we use linear interpolation to estimate the value at 100 hPa. The convective influence ratio is the fraction of the MLS observations that encounter deep convection, as determined by the back trajectory calculations. (B-D) Joint distribution of convection and water vapor time series during 2005-2016 divided into (B) June, (C) July, and (D) August. Solid lines show the linear fit, and dashed lines show the 95% significant level margin of error of the slope bar (accounting for auto-correlation). To account for the time for the water vapor to spread out, each data point is a 10-day average of convection, with water vapor averaged over the last five days of the averaging period for convection.

---

## Author Comment (AC2) · 1 Jul 2020

We thank the reviewer for their useful comments on our manuscript. Below we provide detailed response to your comments. In the following, the comments of the reviewer are presented in *italic blue*. Our responses are in normal black font. Changes to the text are in red.

*Comment 1:*
*The paper presents a hypothesis that deep overshooting convection over the great plains of north America moistens the lower stratosphere and is transported and trapped in the North American anticyclone. they use trajectory calculations to see if water vapor measurements from MLS encountered convection or not. They show that during July and August that the difference between convective and non convective trajectories is nearly 1 ppmv. In June it is much less. The take away message i get from this is that the establishment of the North American anticyclone is the dominant factor for having high lower stratospheric water vapor over North America.*

This is not the impression we were trying to get across. Convection is they key reason that water vapor is higher in the monsoon region. Dynamics and the structure of the monsoon can mediate the impact of convection by, for example, regulating the temperature that air in the monsoon experiences. In fact, one should not really separate these phenomena. After all, it is convection that fundamentally causes the anticyclonic monsoonal circulation. Changes like this will be made in the text:

Line 4 - We use a Lagrangian trajectory model to demonstrate that the structure and the location of the NA anticyclone, as well as the tropical upper tropospheric temperature, mediate the moistening impact of convection.

*It completely dominates convective activity in the Junes of 2010 and 2011 where the former year is high with low NA convection and low the following year despite having more NA convection. I have difficulty believing that in 2010 before the anticyclone is set up that the convectively moistened air moves to the colder tropics where it gets freeze dried first.*

While this might be counterintuitive, we provide evidence in the paper that this is indeed happening. We examine Lagrangian trajectories (e.g., Fig. 6 in the original paper and associated text) and show that parcels are indeed reaching deep into the tropics. If we regard 20°N-20°S as the tropics, then 20% and 44% of the convectively influenced parcels travel into the tropics and back during June 2010 and 2011, respectively, and this difference can lead to significant interannual variability in the monsoon region.

We acknowledge that we should add more numerical information to the paper to make it clear. Changes have been made in the text:

Line 137 - The second reason is also connected to the changing dynamics during June, July, and August. Parcels tend to travel to lower latitudes during June (Fig. 4a). 33% of the convectively moistened parcels travels to the tropics (20°N-20°S) during June, and 13% during July and 9% during August. Traveling to the tropics leads them to experience colder temperatures at 100 hPa (Fig. 4b). As a result, the median of the water vapor mixing ratio of the parcels that stays in the mid-latitudes is 5.98 ppmv, while it is 5.36 ppmv for those parcels that travels to the tropics. This

means that convectively moistened air experiences subsequent dehydration more frequently in June than in later months (Randel et al., 2015).

Line 157 - In June 2010, 20% of the convectively influenced parcels travel to the tropics (20◦N-20◦S) in 5 days, while in June 2011, 44% do (Fig. 6a).

*It seems from looking at the figure 3 and figure 5 that the horizontal tape recorder signature of water transport is playing a big role here too. i certainly agree that deep convection over the NA plains is adding water but it might be a small perturbation on top of the large scale transport coming up from the tropics that is also becoming more moist during the summer months. I think this could be disentangled with some modelling studies where one could artificially hold the tropical tropopause temperature constant all year thus removing the tape recorder signatures from the tropics and seeing what NA enhancements occur just due to local convection.*

There is abundant previously published work that show that the high water vapor found in this region is the result of local convection in the region (Hanisco et al., 2007; Herman et al., 2017; Schwartz et al., 2013; Smith et al., 2017). This can also be seen in Fig. S1, which shows that the water vapor over the monsoon region is a local maximum, meaning that it could not be caused by horizontal transport out of the deep tropics.

To provide further support, Fig. S2 shows the horizontal distribution of parcels initially located in the NA monsoon after 10 day back trajectories. This shows that many of the parcels had been within the NA monsoon for more than 10 days, and those that were not had been transported to the monsoon by westerly winds from the Pacific — not from the tropics.

We will add some discussion to the revised paper discussing how the water vapor maximum in the monsoon is due to local convection:

Line 18 - However, over the Asian monsoon and North American monsoon, higher water vapor mixing ratios, sometimes as high as 12 ppmv, are observed. This value is much higher than the water vapor mixing ratio in the tropics, indicating that the air did not go through the TTL or is moistened further after leaving the TTL (Anderson et al., 2012; Schwartz et al., 2013; Randel et al., 2015; Smith et al., 2017).

Line 80 - We have subtracted the zonal mean value (based on the JJA mean from 70°W to 130°W at each degree of latitude) from NA the 100-hPa water vapor content before normalizing, allowing us to focus on the variability in the NA region relative to the zonal average value and minimize the impact of transportation from the tropics.

*Minor recommendation 1:*
*minor recommendations page 1 line 18 replace sometimes as high as with exceeding*
*... MLS has seen higher values as has Anderson.*

Changes have been made in the text:

Line 18- However, over the Asian monsoon and North American monsoon, higher water vapor mixing ratios, sometimes exceeding 12 ppmv, are observed, indicating that the air did not go through the TTL or is moistened further after leaving the TTL.

*Minor recommendation 2:*
*page 3 line 81 from NA the 100 to from the NA 100...*

Changes have been made in the text:
Line 81 - We have subtracted the zonal mean value (based on the JJA mean from 70°W to 130°W at each degree of latitude) from the NA 100-hPa water vapor content before normalizing, allowing us to focus on the variability in the NA region relative to the zonal average value.

*Minor recommendation 3:*
*Throughout the manuscript lower case letters a, b, .. are used to refer to panels in the figures but the figures use upper case letters A, B, ... Please make this consistent.*

Changes have been made in the text.

*Minor recommendation 4:*
*page 5 line 135 I would write that sentence as As a result moistening from deep convection becomes less diluted by zonal mean flow later in the summer.*

Changes have been made in the text:

As a result, moistening from deep convection becomes less diluted by zonal mean flow later in the summer.

[Figure]

Figure S1. MLS 100-hPa water vapor mixing ratio during 2005-2016 (A) June, (B) July, and (C) August.

[Figure]

Figure S2. Number of the parcels in each 2°*2° grid box after 10 days in the back trajectory model during 2005-2016 (A) June, (B) July and (C) August. In the back trajectory experiments, we initiate the parcels 1°x1° 1∘x 1∘grid over NA (25°N - 50°N, 70°W- 130°W) every day during each month, and track back their position in 10 days. (Black contour) Geographical distribution of the MLS 100-hPa water vapor anomaly (after removal of the zonal mean).

**References**

Hanisco, T. F., Moyer, E. J., Weinstock, E. M., St. Clair, J. M., Sayres, D. S., Smith, J. B., Lockwood, R., Anderson, J. G., Dessler, A. E., Keutsch, F. N., Spackman, J. R., Read, W. G. and Bui, T. P.: Observations of deep convective influence on stratospheric water vapor and its isotopic composition, Geophys. Res. Lett., 34(4), L04814, doi:10.1029/2006GL027899, 2007.

Herman, R. L., Ray, E. A., Rosenlof, K. H., Bedka, K. M., Schwartz, M. J., Read, W. G., Troy, R. F., Chin, K., Christensen, L. E., Fu, D. J., Stachnik, R. A., Bui, T. P. and Dean-Day, J. M.: Enhanced stratospheric water vapor over the summertime continental United States and the role of overshooting convection, Atmos. Chem. Phys., 17(9), 6113–6124, doi:10.5194/acp-17-6113-2017, 2017.

Randel, W. J., Zhang, K. and Fu, R.: What controls stratospheric water vapor in the NH summer monsoon regions?, J. Geophys. Res. Atmos., 120(15), 7988–8001, doi:10.1002/2015JD023622, 2015.

Schwartz, M. J., Read, W. G., Santee, M. L., Livesey, N. J., Froidevaux, L., Lambert, A. and Manney, G. L.: Convectively injected water vapor in the North American summer lowermost stratosphere, Geophys. Res. Lett., 40(10), 2316–2321, doi:10.1002/grl.50421, 2013.

Smith, J. B., Wilmouth, D. M., Bedka, K. M., Bowman, K. P., Homeyer, C. R., Dykema, J. A., Sargent, M. R., Clapp, C. E., Leroy, S. S., Sayres, D. S., Dean-Day, J. M., Paul Bui, T. and Anderson, J. G.: A case study of convectively sourced water vapor observed in the overworld stratosphere over the United States, J. Geophys. Res. Atmos., doi:10.1002/2017JD026831, 2017.

---

## Author Response (AR2)

**Reply to Reviewer 1**

We thank the reviewer for their useful comments on our manuscript. Below we provide detailed response to your comments. In the following, the comments of the reviewer are presented in *italic blue*. Our responses are in normal black font. Changes to the text are in red.

*I request only a relatively minor revision. It is noted that there is interannual variability in H2O (e.g. between June 2010 and June 2011), and Section 5 presents a possible explanation. But it is only a possible and reasonable explanation, not necessarily the only explanation for this interannual difference. I would be satisfied if the authors simply changed in the last sentence of this section from "monsoon dynamics variability plays a significant role" to "monsoon dynamics variability may play a significant role".*

We have made that change.

*Also there is a typo. "moistened parcels travels" should be "moistened parcels travel".*

Corrected.

**List of minor wording changes:**
Line 142: 33% of the convectively moistened parcels  travel to the tropics (20°N-20°S) during June, and 13% during July and 9% during August.

Line 144: As a result, the median of the water vapor mixing ratio of the parcels that  stay in the mid-latitudes is 5.98 ppmv, while 145 it is 5.36 ppmv for those parcels that  travel to the tropics.

Line 167: Thus, monsoon dynamics variability may play a significant role in the generating inter-annual variability of stratospheric water vapor in the NA region.

[revised manuscript text omitted]